# FigGen: Text to Scientific Figure Generation

**Juan A. Rodriguez**[1], **David Vazquez**[1], **Issam Laradji**[1], **Marco Pedersoli**[2] **& Pau Rodriguez**[3],*

[1]ServiceNow Research, [2]LIVIA, ÉTS, Montréal, [3]Apple Machine Learning Research. *Work done at ServiceNow

## Abstract

The generative modeling landscape has experienced tremendous growth in recent years, particularly in generating natural images and art. Recent techniques have shown impressive potential in creating complex visual compositions while delivering impressive realism and quality. However, state-of-the-art methods have been focusing on the narrow domain of natural images, while other distributions remain unexplored. In this paper, we introduce the problem of text-to-figure generation, that is creating scientific figures of papers from text descriptions. We present FigGen, a diffusion-based approach for text-to-figure as well as the main challenges of the proposed task. Code and models are available at https://github.com/joanrod/figure-diffusion

## 1 Introduction

Scientific figure generation is an important aspect of research, as it helps to communicate findings in a concise and accessible way. The automatic generation of figures presents numerous advantages for researchers, such as savings in time and effort by utilizing the generated figures as a starting point, instead of investing resources in designing figures from scratch. Making visually appealing and understandable diagrams would allow accessibility for a wider audience. Furthermore, exploring the generative capabilities of models in the domain of discrete graphics would be of high interest.

Generating figures can be a challenging task, as it involves representing complex relationships between discrete components such as boxes, arrows, and text, to name a few. Unlike natural images, concepts inside figures may have diverse representations and require a fine-grained understanding. For instance, generating a diagram of a neural network presents an ill-posed problem with high variance, as it can be represented by a simple box or an unfolded representation of its internal structure. Human understanding of figures largely relies on the text rendered within the image, as well as the support of text explanations from the paper written in technical language.

By training a generative model on a large dataset of paper-figure pairs, we aim to capture the relationships between the components of a figure and the corresponding text in the paper. Dealing with variable lengths and highly technical text descriptions, different diagram styles, image aspect ratios, and text rendering fonts, sizes, and orientations are some of the challenges of this problem. Inspired by impressive results in text-to-image, we explore diffusion models to generate scientific figures. Our contributions are i) introduce the task of text-to-figure generation and ii) propose FigGen, a latent diffusion model that generates scientific figures from text captions.

**Related work.** Deep learning has emerged as a powerful tool for conditional image generation (Ramesh et al., 2022; Saharia et al., 2022; Balaji et al., 2022), thanks to advances in techniques such as GANs (Goodfellow et al., 2014; Karras et al., 2019; 2021) and Diffusion (Sohl-Dickstein et al., 2015; Ho et al., 2020; Song et al., 2020b). In the domain of scientific figures, Rodriguez et al. (2023) presented Paper2Fig100k, a large dataset of paper-figure pairs. In this work, we aim to explore diffusion models applied to the task of text-to-figure generation and analyze its challenges.

## 2 Method and Experiments

We train a latent diffusion model (Rombach et al., 2021) from scratch. First, we learn an image autoencoder that projects images into a compressed latent representation. The image encoder uses a KL loss and OCR perceptual loss (Rodriguez et al., 2023). The text encoder used for conditioning is

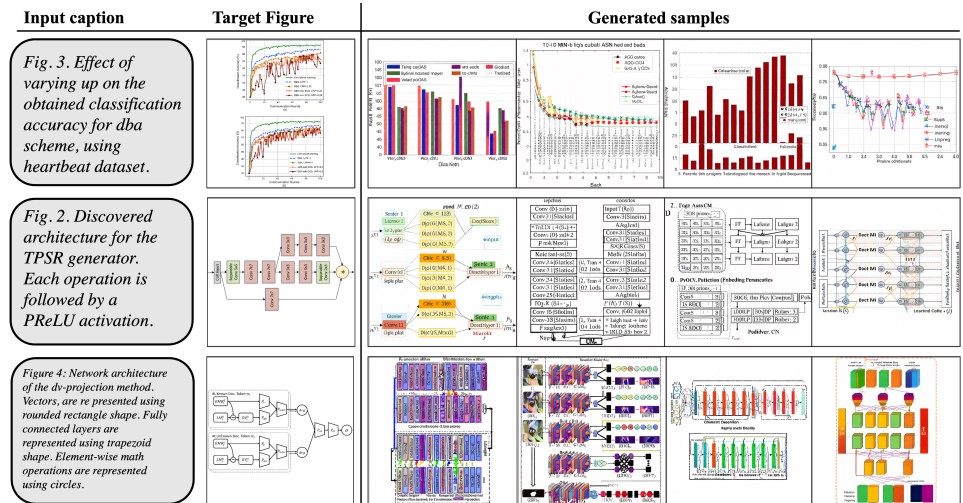

Figure 1: Samples generated by our model using captions from Paper2Fig100k test set.

| Model | Text encoder | Parameters | CFG | FID↓ | IS↑ | KID↓ | OCR-SIM↓ |
|---|---|---|---|---|---|---|---|
| FigGen$_{Base}$ | Bert (8 layers) | 866M | 1.0 | 302.46 | 1.04 | 0.32 | 5.97 |
| FigGen$_{Base}$ | Bert (8 layers) | 866M | 5.0 | 282.32 | **1.09** | **0.29** | 5.89 |
| FigGen$_{Base}$ | Bert (8 layers) | 866M | 10.0 | 284.12 | 1.08 | **0.29** | 5.83 |
| FigGen$_{Mid}$ | Bert (32 layers) | 942M | 1.0 | 308.50 | 1.03 | 0.32 | 5.95 |
| FigGen$_{Mid}$ | Bert (32 layers) | 942M | 5.0 | 298.98 | 1.06 | 0.31 | 5.91 |
| FigGen$_{Mid}$ | Bert (32 layers) | 942M | 10.0 | 301.10 | 1.06 | 0.31 | 5.86 |
| FigGen$_{Large}$ | Bert (128 layers) | 1.2B | 1.0 | 302.99 | 1.04 | 0.32 | 6.08 |
| FigGen$_{Large}$ | Bert (128 layers) | 1.2B | 5.0 | **281.25** | **1.09** | **0.29** | **5.74** |
| FigGen$_{Large}$ | Bert (128 layers) | 1.2B | 10.0 | 288.02 | **1.09** | **0.29** | 5.76 |

Table 1: Main quantitative results of our text to figure generation models.

learned end-to-end during the training of the diffusion model. The diffusion model interacts directly in the latent space and performs a forward schedule of data corruption while simultaneously learning to revert the process through a time and text conditional denoising U-Net (Ronneberger et al., 2015) (see Appendix A.1 for details). We use Paper2Fig100k, composed of figure-text pairs from research papers. It consists of $81,194$ samples for training and $21,259$ for validation.

**Experimental results.** During generation, we use DDIM (Song et al., 2020a) sampler with 200 steps and generate $12,000$ samples for each model to compute FID, IS, KID (Regenwetter et al., 2023), and OCR-SIM[1]. We use classifier-free guidance (CFG) to test super-conditioning (Ho & Salimans, 2022). Table 1 presents results of different text encoders, and Figure 1 shows generated samples of FigGen$_{Base}$. We find that the large text encoder offers the best results and that we can improve conditional generation by increasing the CFG scale. Although qualitative samples do not present sufficient quality to solve the task, FigGen has learned interesting relationships between texts and figures such as the difference between plots and architectures (see also Appendix A.3).

## 3 CONCLUSION

In this paper, we introduce the task of text-to-figure generation and define FigGen, a latent diffusion model that we train on the Paper2Fig100k dataset. Our experiments show that FigGen is able to learn relationships between figures and texts and generate images that fit the distribution. However, these generations are not ready to be useful for researchers. One of the main challenges to solve is the variability in text and images, and how to better align both modalities. Also, future work must design validation metrics and loss functions for generative models of discrete objects.

---

[1]https://github.com/joanrod/ocr-vqgan

**Ethics statement.** A central concern of this work is fake paper generation. To address this, we consider building classifiers or using watermarks for the detection of fake content. However, further research is needed to elucidate how these systems should be made public.

**URM Statement.** The authors acknowledge that at least one key author of this work meets the URM criteria of ICLR 2023 Tiny Papers Track.

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

| | |
|---|---|
| Input (latent) shape | 64 x 64 x 4 |
| Number of channels | 256 |
| Number of residual blocks | 3 |
| Self-attention resolutions | [64, 32, 16] |
| Channel mult. | [1, 2, 4, 4] |
| Dropout | 0 |

Table 2: Base diffusion U-Net architecture.

Jascha Sohl-Dickstein, Eric A. Weiss, Niru Maheswaranathan, and Surya Ganguli. Deep unsupervised learning using nonequilibrium thermodynamics. *arXiv*, abs/1503.03585, 2015.

Jiaming Song, Chenlin Meng, and Stefano Ermon. Denoising diffusion implicit models. *arXiv preprint arXiv:2010.02502*, 2020a.

Yang Song, Jascha Sohl-Dickstein, Diederik P Kingma, Abhishek Kumar, Stefano Ermon, and Ben Poole. Score-based generative modeling through stochastic differential equations. *arXiv preprint arXiv:2011.13456*, 2020b.

Richard Zhang, Phillip Isola, Alexei A. Efros, Eli Shechtman, and Oliver Wang. The unreasonable effectiveness of deep features as a perceptual metric. *arXiv*, abs/1801.03924, 2018.

## A    APPENDIX

### A.1    MODEL DETAILS

**Image encoder.** The first stage, the image autoencoder, is devoted to learning a projection from the pixel space to a compressed latent representation that makes the diffusion model train faster. The image encoder needs to also learn to project the latents back to the pixel space without losing important details about the figure (e.g., text rendering quality). To this end, we define a convolutional encoder and decoder with a bottleneck, that downsamples images with a factor $f = 8$. The encoder is trained to minimize a KL loss with a gaussian distribution as well as a VGG perceptual (Zhang et al., 2018) loss and an OCR perceptual loss. We follow the adversarial procedure proposed in VQGAN (Esser et al., 2020), which increases reconstruction quality.

**Text encoder.** We find that using a general-purpose text encoder (e.g., CLIP (Radford et al., 2021)) is not well-suited for our task, because text encoders trained on natural texts and images exhibit a domain gap with respect to the technical descriptions present in papers. We define Bert (Devlin et al., 2018) transformer that is trained from scratch during the diffusion process. We use an embedding channel of size $512$, which is also the embedding size of the cross-attention layers for conditioning the U-Net. We explore varying the number of layers of the transformer in the set $8, 32, and 128$.

**Latent diffusion model.** Table 2 describes the U-Net network architecture. We perform diffusion in a perceptually equivalent latent representation of images, that is compressed to an input size of $64x64x4$, which makes the diffusion model faster. We define 1000 steps of diffusion and a linear noise schedule.

### A.2    TRAINING DETAILS

Our models are trained in Paper2Fig100k. During our experiments, we find that a challenging problem is how to deal with extremely varying aspect ratios that exist between images in the dataset (e.g., figures tend to be larger in width than in height). Cropping figures would result in a loss of crucial information. Therefore we opt for applying white padding to images and only consider images with an aspect ratio between $0.5$ and $2$, to avoid having most of the pixel information be the padded pixels. The final dataset consists of $37,613$ samples for training and $9,506$ for validation. Images are processed at size $512x512$ and downsampled to $64x64$ by our image autoencoder.

For training the image autoencoder we use Adam optimizer with an effective batch size of 4 samples, and a learning rate of $4.5e-6$, using 4 Nvidia V100 12GB GPUs. For training stability, we warm up

| | |
|---|---|
| Input (image) shape | 384 x 384 x 3 |
| Embed dimension | 4 |
| Number of channels | 128 |
| Number of residual blocks | 2 |
| Channel mult. | [1, 2, 4, 4] |
| Discriminator weight | 0.5 |
| VGG perceptual loss weight | 0.2 |
| OCR perceptual loss weight | 0.8 |
| KL loss weight | 1e-6 |
| Dropout | 0 |

Table 3: Image autoencoder architecture.

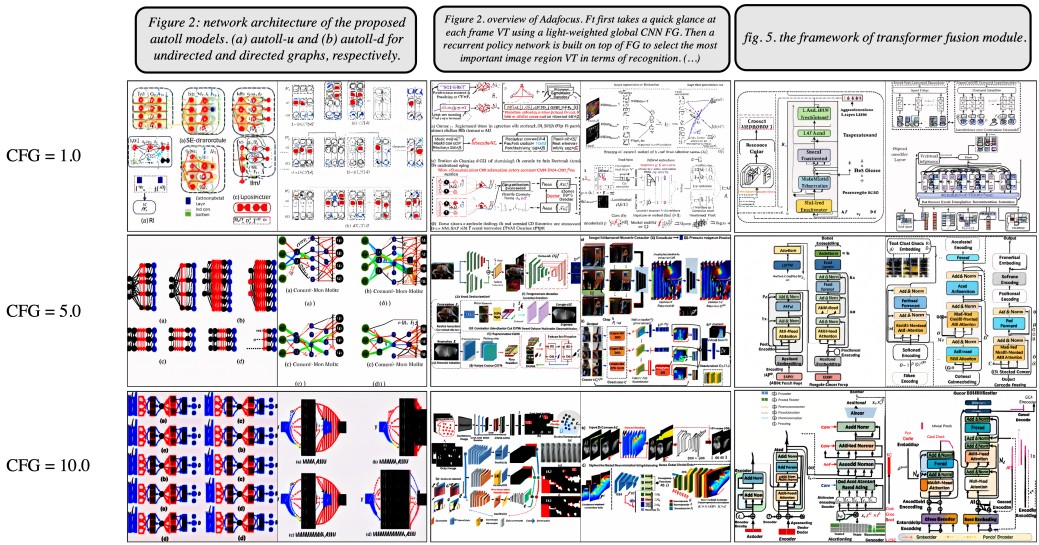

Figure 2: Samples generated from FigGen$_{\textbf{Base}}$. We display two generated samples for three prompts Paper2Fig100k test set. Each row displays the classifier-free guidance (CFG) scale, showing that we can super-condition the generations to make the samples align better with the prompt.

the model during $50k$ iterations without using the discriminator (Esser et al., 2020; Rombach et al., 2021). For training latent diffusion models, we use Adam optimizer, using an effective batch size of 32 and a learning rate of $1e - 4$. For training the models on Paper2Fig100k we use 8 Nvidia A100 80GB GPU.

## A.3 ADDITIONAL GENERATION RESULTS

Figures 2 show additional generated samples of FigGen when tuning the classifier-free guidance (CFG) (Ho & Salimans, 2022) parameter. We observe improvement in figure quality when increasing the CFG scale which is also shown quantitatively. Figure 3 presents more generations of FigGen. Note the variability in text length between samples, as well as the technical level of the captions, which makes it difficult for the model to properly generate understandable figures. However, high-level concepts are correctly captured, such as captions that describe charts and plots, algorithms, or cases where we aim to display neural network architectures.

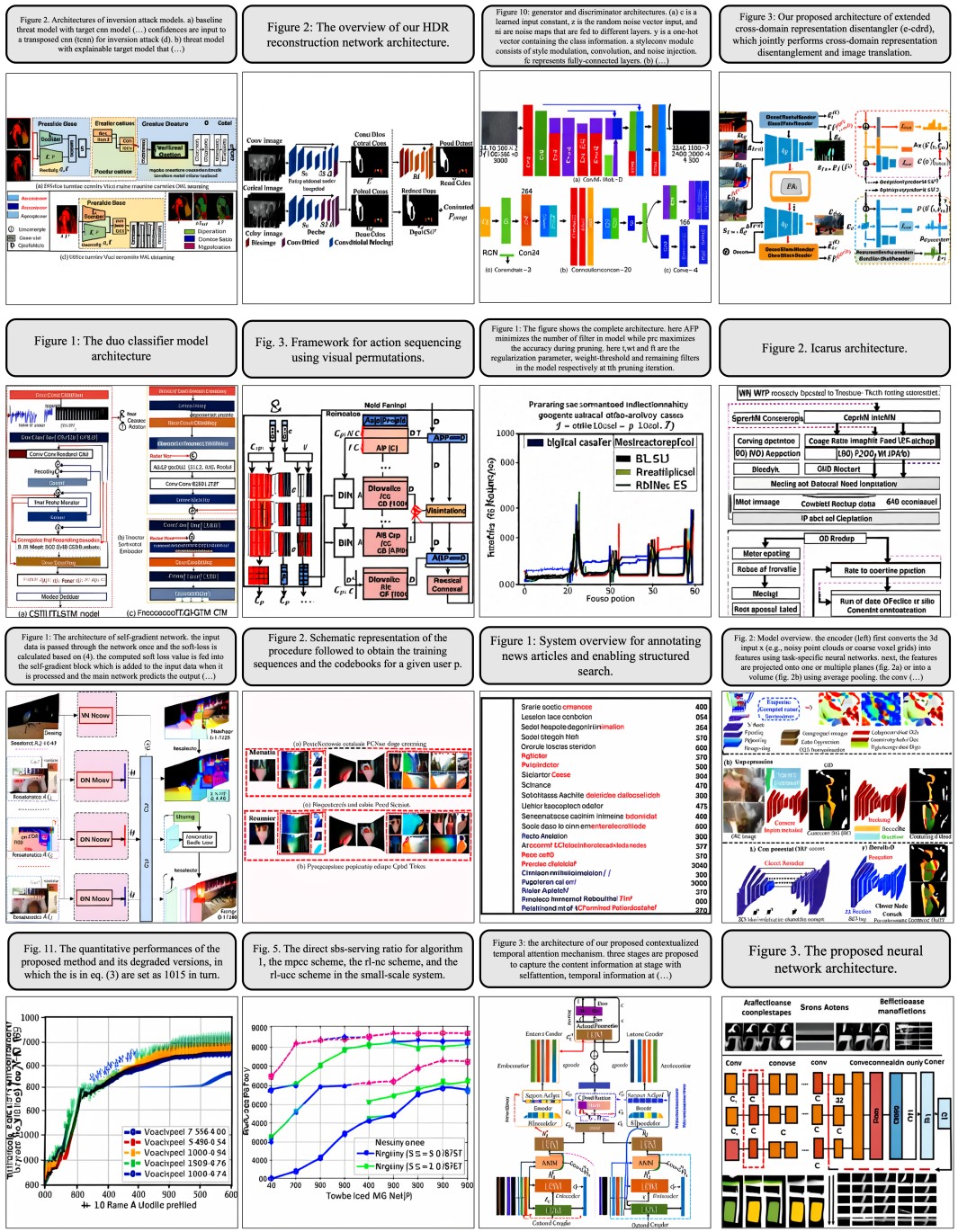

Figure 3: Generated samples from the test set prompts above as input. Samples generated using FigGen**Base**.

