# OpenReview forum: "FigGen: Text to Scientific Figure Generation"
_ICLR.cc/2023/TinyPapers — Submitted to Tiny Papers @ ICLR 2023_

### Official Review · Reviewer_jXn8 · 2023-03-30

**Confidence:** 3

**Summary Of Contributions:**

This paper introduces the task of text-to-figure generation, which involves creating scientific figures for papers based on textual descriptions. The authors propose a diffusion-based approach to tackle this problem and discuss some challenges.

**Rating:**

Great Start (GS): a submission which meets some of the reviewing criteria but has room for improvement

**Strengths And Weaknesses:**

Strengths:
1. The paper presents a diffusion-based approach for the text-to-figure generation task, showcasing its applicability in this domain.
2. The authors conduct extensive experiments, evaluating the performance of various text encoders and different scales of the CFG.
3. Code and models will be made public.

Weaknesses:
1. The core findings of the paper are not clearly presented. It would have been helpful if the authors had provided a comparison between existing models.
2. The paper would benefit from additional evaluation metrics, such as OCR-SIM. The FID metric is primarily statistical, and considering the importance of text in scientific figures, incorporating more text-centric metrics would provide a more comprehensive evaluation.

**Suggested Changes:**

See weakness.
Typos:
1. In Table 1, "ISC" or "IS"?

---

> ### Author Response · Authors · 2023-06-01
> **Comment on the review**
>
> Dear reviewer,
>
> Thank you for the constructive feedback and for recognizing our diffusion-based approach, extensive experimentation, and commitment to open science. Regarding your concerns:
>
> Clarity of core findings: We understand the need for a clearer presentation of our findings. As we present the first work on the text-to-figure generation task, we recognize the need to better emphasize the novelty and impact of our work. We will make these aspects more prominent in our revision
>
> Additional evaluation metrics: We agree with your suggestion to include OCR-SIM, therefore, in our revised manuscript, we also report OCR-SIM.
>
> Typographical errors: We appreciate your observation, and correct the "ISC" error in Table 1 in our revision.
>
> Your constructive comments and rating are valuable as they guide our improvements, and we believe that the revised paper will be significantly enhanced. Thank you again for your time and insightful suggestions.

---

### Comment · Area_Chair_bx8r · 2023-06-06
**Archival**

 This work meets the threshold for archival, contains the URM statement (without specifying the author who meets the requirements) and is deanonymized.

---

### Meta-Review · Area_Chair_bx8r · 2023-04-06

**Recommendation:** Invite to present
**Confidence:** 4

**Metareview:**

- Clarity: The paper is clear, well-written, provides a literature context, and points to future directions and ethical concerns.
- Correctness: Experiments are satisfactory and provide several quantitative and qualitative analyses. The only drawback is missing of competitors, which does not make clear how hard the proposed task is and what the positioning of the method among other alternatives
- Reproducibility: The paper provides details of the used architecture and promises to release the code. Hence, I consider the experiments well reproducible.

The only review collected appreciated the clarity, the experiments, and the promised code, but also pointed out the possibility of extending the experimental evaluation with more metrics and competitors and asked to elaborate more on the paper's findings.

**Summary:**

The paper proposes a diffusion-based method to generate scientific figures from text input. The paper reaches the bars of Clarity, Correctness, and Reproducibility, while some tuning (metrics, competitors, discussion of the results) could be beneficial.

**Comments And Feedback To The Authors:**

I enjoyed the proposed problem, and the results look pretty promising. I suggest considering compression with baselines, which seems to be the main drawback. This would strengthen the message, pointing the reader to what can be done in future. Figures are a bit tiny, in my opinion: it would be nice to have in the appendix some large images to point out the quality of the result better. It would be interesting to discuss recurring issues in the qualitative results and what is harder to achieve for the proposed method. I also think that probably an analysis of different complexities in the text prompt could be curious, especially focusing on some specific properties of the desired figure (e.g., "a bar plot, with three bins" vs. "a bar plot, with three bins of different colours" vs. "a bar plot, with three bins where colours are colourblind-friendly").

**Reason For Not Giving A Higher Recommendation:**

The lack of baselines limits the possibility of discussing the impact of the proposed methodology. This would also help to better discuss the outcome of the approach and sharpen the future possible methodological directions.

**Reason For Not Giving A Lower Recommendation:**

The problem is interesting, and the work succeeds in presenting the approach clearly, correctly, and in a reproducible way. While not directly useful, the paper can foster discussion and seems a nice fit for the ICLR community.

---

### Decision · Program_Chairs · 2023-04-07

Invite to present